# Response Predictive Markers and Synergistic Agents for Drug Repositioning of Statins in Ovarian Cancer

**DOI:** 10.3390/ph15020124

**Published:** 2022-01-21

**Authors:** Yusuke Kobayashi, Takashi Takeda, Haruko Kunitomi, Fumiko Chiwaki, Masayuki Komatsu, Shimpei Nagai, Yuya Nogami, Kosuke Tsuji, Kenta Masuda, Hideaki Ogiwara, Hiroki Sasaki, Kouji Banno, Daisuke Aoki

**Affiliations:** 1Department of Obstetrics and Gynecology, Keio University School of Medicine, 35 Shinanomachi, Shinjuku-ku, Tokyo 160-8582, Japan; double.tak777@keio.jp (T.T.); harukoirie1225@gmail.com (H.K.); nagai19851126@keio.jp (S.N.); y-nogami.a7@keio.jp (Y.N.); kosuke_tsuji@a3.keio.jp (K.T.); ma-su-ken@a2.keio.jp (K.M.); kbanno@z7.keio.jp (K.B.); aoki@z7.keio.jp (D.A.); 2Department of Translational Oncology, National Cancer Center Research Institute, 5-1-1 Tsukiji, Chuo-ku, Tokyo 104-0045, Japan; fchiwaki@ncc.go.jp (F.C.); makomats@ncc.go.jp (M.K.); hksasaki@ncc.go.jp (H.S.); 3Division of Cancer Therapeutics, National Cancer Center Research Institute, 5-1-1 Tsukiji, Chuo-ku, Tokyo 104-0045, Japan; hogiwara@ncc.go.jp

**Keywords:** ovarian cancer, drug repurposing, statin, *VDAC1*, *LDLRAP1*

## Abstract

In the field of drug repurposing, the use of statins for treating dyslipidemia is considered promising in ovarian cancer treatment based on epidemiological studies and basic research findings. Biomarkers should be established to identify patients who will respond to statin treatment to achieve clinical application. In the present study, we demonstrated that statins have a multifaceted mode of action in ovarian cancer and involve pathways other than protein prenylation. To identify biomarkers that predict the response to statins, we subjected ovarian cancer cells to microarray analysis and calculated Pearson’s correlation coefficients between gene expression and cell survival after statin treatment. The results showed that *VDAC1* and *LDLRAP1* were positively and negatively correlated with the response to statins, respectively. Histoculture drug response assays revealed that statins were effective in clinical samples. We also confirmed the synergistic effects of statins with paclitaxel and panobinostat and determined that statins are hematologically safe to administer to statin-treated mice. Future clinical trials based on the expression of the biomarkers identified in this study for repurposing statins for ovarian cancer treatment are warranted.

## 1. Introduction

The recent emergence of drugs that target specific molecules and antibodies for ovarian cancer treatment has immensely improved the therapeutic outcomes [1]; however, a paradigm shift from treatment to prevention is required in the future. The development of new drugs requires long-term, multifaceted basic research, but the resulting drugs are expensive owing to the large amount of money spent on their development. The probability of drug approval from the seeds is extremely low; thus, novel drug discovery methods are needed [2,3]. One solution to this problem is drug repurposing. It involves the use of an existing drug for a disease for which it was not developed. The safety of these drugs has already been confirmed in humans. This approach has attracted considerable attention in recent years as it enables low-cost and highly reliable drug discovery [3,4,5,6]. As a drug repurposed for ovarian cancer, statins, which are used to treat dyslipidemia, have been investigated in epidemiological studies, basic research, and clinical trials [7]. Statins, used worldwide for dyslipidemia treatment, reduce blood cholesterol levels by inhibiting hydroxy methylglutaryl coenzyme A (HMG-CoA) reductase, which is located upstream of the mevalonate pathway that biosynthesizes cholesterol from acetyl coenzyme A [8]. As the mevalonate pathway is also involved in the prenylation of the cancer-associated proteins Ras and Rho, statins, which inhibit this pathway, are expected to have antitumor effects [9]. A reduction in the risk of cancer-related mortality in statin users has been reported in 13 cancer types [10], and epidemiological studies on cancers, including colorectal cancer, prostate cancer, and esophageal cancer, have reported the antitumor effects of statins [11,12,13]. Many reports on the antitumor effects of statins have resulted from basic research, and we have also reported the antitumor effects of statins on ovarian cancer both in vitro and in vivo [14,15]. However, clinical trials on the drug repositioning of statins for ovarian cancer have not yet shown promising results; thus, the design of trials will need to be improved further [16]. To conduct an efficient clinical trial, in this study, we aimed to understand the details of the mode of action of statins, establish biomarkers to select patients who would respond to statins, confirm that statins would respond to clinical samples, ensure the safety of administration, and examine the synergistic effects of combination therapy to be tested in clinical trials.

## 2. Results

### 2.1. Mode of Action of Statins Other Than Protein Prenylation

We previously showed that statins have a proliferative and tumor-suppressive effect on ovarian cancer cells by inhibiting the mevalonate pathway, particularly the farnesylation and geranylgeranylation pathways, which are involved in protein prenylation [14]. However, statins have a multifaceted mode of action [17]. To investigate whether protein prenylation is the only pathway and mode of action of statins, we compared the inhibitory effect of L-778123 [18], a simultaneous inhibitor of both farnesyltransferase and geranylgeranyltransferase, and statins on cell proliferation. The IC50 of simvastatin was 2.99 μM for OVSAHO cells and 20.5 μM for KURAMOCHI cells, and the IC50 of L778123 was 56.9 μM for OVSAHO cells and >100 μM for KURAMOCHI cells, indicating that simvastatin has a lower IC_50_ value than L-778123. This observation raises one possibility that statins may have other modes of action besides protein prenylation (Figure 1a). To extract the genes and pathways regulated by simvastatin compared with those regulated by L-778123, we performed microarray analysis using OVSAHO and KURAMOCHI cells cultured with simvastatin and L-778123, respectively (GSE183473; https://www.ncbi.nlm.nih.gov/geo/query/acc.cgi?acc=GSE183473, accessed on 10 November 2021). In comparison with L-778123, genes significantly regulated by simvastatin in simvastatin-responsive OVSAHO cells were extracted under the condition that the fold change in expression between OVSAHO and KURAMOCHI cells cultured with simvastatin was 0.800–1.200 (Appendix A; see Appendix A). Genes for which the expression was upregulated by simvastatin compared with those upregulated by L-778123 included *PIK3IP1*, *BMF*, *ATF3*, *MGP*, and *ERVH-3*, whereas genes for which expression was downregulated included *GSTA9P*, *MID1*, *ERVMER34-1*, *ZG16*, and *RABGEF1*. Gene ontology (GO) analysis showed that the biological processes that were significantly upregulated by simvastatin included cell cycle process, chromosome segregation, and nuclear chromosome segregation; GO cellular components included chromosome, nonmembrane-bounded organelle, and kinetochore; and GO molecular functions included catalytic activity, acting on DNA, actin binding, and cytoskeletal protein binding (Appendix A; see Appendix A). GO biological processes that were significantly downregulated by simvastatin compared with those downregulated by L-778123 included the positive regulation of release of cytochrome c from mitochondria, apoptotic signaling pathway, and cell aggregation; GO cellular components included the integral component of plasma membrane, extracellular space, and intrinsic component of plasma membrane; and GO molecular functions included spermidine binding and diamine N-acetyltransferase activity (Appendix A; see Appendix A). Certain morphological changes, such as fragmentation and vacuole formation, were observed in ovarian cancer cells after the addition of lovastatin (Figure 1b), suggesting that statins induce programmed cell death processes such as apoptosis and autophagy. The activity of caspase3, a marker of apoptosis, was measured using the luciferase assay, and it was found to significantly increase in a concentration-dependent manner with the addition of lovastatin (Figure 1c). The expression of LC3, an autophagy marker, was quantitatively confirmed using fluorescent DsRed-labeled plasmids, and the formation of autophagosomes expressing LC3 was significantly increased in cells treated with lovastatin (Figure 1d). Statins, therefore, induced apoptosis and autophagy, suggesting that these mechanisms of programmed cell death are involved in the antitumor effects of statins. Taken together, these results indicate that statins exert their antitumor effects not only by inhibiting protein prenylation but also via multiple pathways and have a multifaceted mode of action.

### 2.2. Identification of Biomarkers That Predict the Response to Statins

To identify biomarkers that could be used to predict the response to statins, the involvement of histopathological subtypes was investigated. The cell viability of nine ovarian cancer cell lines, i.e., A2780, ES-2, JHOM1, MCAS, OVISE, OVSAHO, RMG-1, RUMG-S, and TOV-21G, in simvastatin-treated culture differed between response-prone and nonresponse-prone cell lines (Figure 2a). The classification of these cell lines as per histological type based on previous reports [19,20] revealed that serous and clear cell carcinoma cell lines were particularly responsive to simvastatin (Figure 2a). The cell viability of cell lines established from human ovarian cancer ascites (NOVC-1C, 2C, 4C, 5C, 7C, and 8C) under simvastatin-treated culture showed that there was a difference in response among the cell lines, which ranged from response-prone cell lines (NOVC-4C and 7C) to response-resistant cell lines (NOVC-2C and 8C) (Figure 2b). Hence, to identify the genes involved in the response to simvastatin, 15 cell lines, i.e., 9 existing ovarian cancer cell lines and 6 human ovarian cancer ascites-derived cell lines, were classified into response and nonresponse groups based on cell viability after simvastatin treatment. Pearson’s product–moment correlation coefficient was calculated from gene expression in the microarray analysis and cell viability, which was measured after simvastatin treatment (Table 1). We focused on *VDAC1* as the gene positively correlated with simvastatin response and *LDLRAP1* as the gene negatively correlated with simvastatin response. Because *VDAC1* is involved in mitochondrial energy regulation [21], we suspected that it might contribute to the phenomenon of interference with the Warburg effect of statins, which we previously reported [15,22]. Although *LDLRAP1* is involved in hypercholesterolemia [23], we focused on this gene because statins are involved in the regulation of blood cholesterol levels by inhibiting the mevalonate pathway [14], and a significant correlation has been reported between ovarian tumors and cholesterol levels [24].

*VDAC1* was highly expressed in simvastatin-responsive cell lines, whereas its expression was decreased in cell lines that did not respond well to simvastatin. By contrast, *LDLRAP1* expression was low in simvastatin-responsive cell lines, whereas its expression was upregulated in simvastatin-refractory cell lines (Figure 3a). To confirm this result, we evaluated *VDAC1* and *LDLRAP1* expression in each of the existing cell lines and human ovarian cancer ascites-derived cell lines. We found that *VDAC1* was highly expressed in the serous and clear cell carcinoma cell lines (Figure 3b) and in the simvastatin-responsive cell lines 4C and 7C (Figure 3b), whereas it was expressed at low levels in the mucinous carcinoma cell lines (Figure 3b) and in the simvastatin-refractory cell lines 2C and 8C (Figure 3b). *LDLRAP1* expression was low in the serous and clear cell carcinoma cell lines (Figure 3c) and in the simvastatin-responsive cell lines 4C and 7C (Figure 3c) and high in the mucinous carcinoma cell lines (Figure 3c) and in the simvastatin-refractory cell lines 2C and 8C (Figure 3c). In summary, *VDAC1* and *LDLRAP1* expression could be a predictive marker of statin response.

### 2.3. Confirmation of the Response to Statins in Clinical Samples, Their Safety in Mice, and Evaluation of Drug Combination Effects

Having identified potential biomarkers that will be important to translate into clinical use, we next tested the response to statins in clinical samples. Clinical specimens were incubated with a statin and with paclitaxel and carboplatin (for comparison), and a histoculture drug response assay (HDRA), which is considered to be highly correlated with clinical efficacy, was performed. The inhibition index, which reflects the percentage of growth inhibition, was calculated by measuring the absorbance using the MTT assay. The clinical information of 20 patients is listed in Table 2. The median inhibition index with simvastatin was 49.1% (range, 6.0–77.7%), with paclitaxel (40 μg/mL) was 74.4% (34.2–86.3%), and with carboplatin (30 μg/mL) was 36.1% (11.2–62.9%; Table 2). The inhibition index of simvastatin was lower than that of paclitaxel but higher than that of carboplatin, and the response of simvastatin was also demonstrated in clinical samples. In terms of histology, simvastatin responded to serous and clear cell carcinoma, which had also responded in vitro, as well as to endometrioid carcinoma (Table 2).

We next investigated the safety of administration, a problem in actual clinical practice. With respect to the safety of statin administration, a previous report demonstrated that there was no change in the body weight of mice during statin administration [14]. In addition to these external changes, in the present study, blood samples from lovastatin-treated mice and their controls were used to measure hemograms and clinical biochemistry and thereby determine the drug toxicity of statin. There was no obvious toxicity in the hematology profile (Figure 4a) and clinical chemistry (Figure 4b) of statin and the control, regardless of the method of administration, indicating the safety of statin administration. Finally, we examined the efficacy of combination therapy with other drugs, which may be used in clinical treatment for ovarian cancer. AZD8055 targeting mTOR, copanlisib targeting PI3K, dabrafenib targeting RAF, doxorubicin targeting topoisomerase II, etoposide targeting topoisomerase Ⅱ, irinotecan targeting topoisomerase Ⅰ, niraparib targeting PARP, paclitaxel targeting tubulin, panobinostat targeting HDAC, and trametinib targeting MEK were used in a SynergyFinder™ study to identify and compare their synergistic effects with statins (Table 3). We previously reported that statins may suppress glutathione and avoid the Warburg effect [15], whereas clear cell carcinoma is resistant to anticancer drugs via the enhancement of the Warburg effect using glutathione [25]. Thus, in this study, we evaluated the combination index (CI) of each drug in ES-2 cells, which had the highest response to statins among the clear cell carcinoma cell lines. CI was calculated as CI_50_ for 50% viability, CI_75_ for 25% viability, and CI_90_ for 10% viability. The CIs (average, standard deviation) for simvastatin and each drug were as follows: AZD8055 CI_50_ (0.95, 0.35), CI_75_ (0.87, 0.22), and CI_90_ (>1.30, NA); copanlisib CI_50_ (0.99, 0.08), CI_75_ (0.85, 0.29), and CI_90_ (>1.84, NA); dabrafenib CI_50_ (0.80, 0.21), CI_75_ (>1.05, NA), and CI_90_ (>1.58, NA); doxorubicin CI_50_ (1.27, 0.40), CI_75_ (1.20, 0.24), and CI_90_ (1.05, 0.08); etoposide CI_50_ (1.19, 0.23), CI_75_ (1.07, 0.10), and CI_90_ (0.89, 0.10); irinotecan CI_50_ (1.27, 0.18), CI_75_ (1.04, 0.08), and CI_90_ (0.84, 0.09); niraparib CI_50_ (1.10, 0.19), CI_75_ (1.02, 0.19), and CI_90_ (0.80, 0.08); paclitaxel CI_50_ (0.79, 0.20), CI_75_ (1.05, 0.34), and CI_90_ (0.34, 0.19); panobinostat CI_50_ (0.92, 0.30), CI_75_ (0.56, 0.25), and CI_90_ (>0.88, NA); and trametinib CI_50_ (1.11, 0.20), CI_75_ (1.10, 0.24), and CI_90_ (1.11, 0.36). The combination of simvastatin and paclitaxel was strongly synergistic at all examined ratios, with high efficacy (Figure 5a). The combination of simvastatin and panobinostat was also synergistic at high percentages and was strongly synergistic with higher effect concentrations at the ratios of 1:1 and 1:4 (Figure 5b). Having confirmed the response of statins in clinical specimens and the safety of statin administration in mice and having identified candidate anticancer drugs with potential synergistic effects, we applied statins as repurposed drugs for ovarian cancer therapy.

## 3. Discussion

Many epidemiological studies and meta-analyses have reported on the value of statins for ovarian cancer. They generally suggest that statins are useful but call for further prospective randomized controlled trials [26,27]. Research has reported no causal relationship between statins and the incidence or mortality of ovarian cancer, and no consensus on this topic has been achieved [28]. One study examined the risk of developing epithelial ovarian cancer in patients with genetically proxied inhibition of HMG-CoA reductase caused by a single nucleotide polymorphism in a gene associated with reduced functionality of HMG-CoA reductase. The study found that the genetically proxied inhibition of HMG-CoA reductase, which is equivalent to reducing the level of LDL cholesterol by 1 mmol/L, was associated with a 40% lower risk of developing epithelial ovarian cancer (odds ratio, 0.60; 95% confidence interval, 0.43–0.83; *p* = 0.002), and cases with BRCA1/2 variants also had a 31% lower risk of developing epithelial ovarian cancer (hazard ratio, 0.69; confidence interval, 0.51–0.93; *p* = 0.01) [29]. This study is extremely interesting because it examined the contentious relationship between statins and inhibition of ovarian cancer at the genetic level, and the results strongly suggest that statins that pharmacologically inhibit HMG-CoA reductase have the same effect. No clinical trials have reported that statin administration suppresses the risk of developing ovarian cancer, and it is necessary to investigate for clinicopathological and molecular biological factors that can predict the response to statins to narrow down the candidate patients who will respond to statins. In the present study, we first examined the histopathological types as clinical biomarkers. Based on the cell viability of nine ovarian cancer cell lines (A2780, ES-2, JHOM1, MCAS, OVISE, OVSAHO, RMG-1, RUMG-S, and TOV-21G) treated with statins, serous carcinoma and clear cell carcinoma are expected to be the most likely histological types to respond to statins. However, the results of the HDRA assay on clinical specimens suggested that not all serous and clear cell carcinomas respond to HDRA, although the number of studies is still too small to be conclusive. Molecular biological factors, rather than histopathological factors, should be the focus of attention. Hence, we identified correlated genes by calculating Pearson’s product–rate correlation coefficient from the cell viability of ovarian cancer cell lines and human ovarian cancer ascites-derived cell lines treated with statins. Although a further evaluation of clinical samples is required, *VDAC1* was identified in this study as a gene associated with statin response. *VDAC1* encodes a channel protein of approximately 30 kDa that is found in the outer membrane of mitochondria [30], and three isoforms, *VDAC1*, *VDAC2*, and *VDAC3*, have been identified in mammals [31,32,33,34]. As VDAC can nonselectively permeate substances with a molecular weight of approximately ≤6000 Da, it can transport substrates necessary for energy metabolism from the cytoplasm to the intermembrane region and metabolites from the intermembrane region to the cytoplasm; therefore, it is an important protein for efficient energy metabolism in the mitochondria [35]. VDAC plays a role in the interconnection between the regulation of glycolysis and mitochondrial respiration by binding to hexokinase, the rate-limiting enzyme of glycolysis [36,37]. *VDAC1* expression is upregulated in numerous human cancer cell lines compared with that in normal cell lines, and VDAC1 is therefore a potential therapeutic target in cancer [36,38]. We previously reported the multifaceted effects of dyslipidemic statins on ovarian cancer both in vitro and in vivo, including cell growth inhibition and antitumor effects [14], and we also reported that statins shift the energy production of ovarian cancer cells from the glycolytic system by the cancer-specific Warburg effect to oxidative phosphorylation in mitochondria via activation of the TCA cycle [15]. Coincidentally, *VDAC1*, which is involved in mitochondrial energy regulation, was shown to be involved in the response to statins in this study. In addition, VDAC1 regulates metabolites, ions, and reactive oxygen species, which are highly associated with ovarian clear cell carcinoma [39]. Because alterations in the PI3K/AKT/mTOR pathway are often found in ovarian clear cell carcinoma, this pathway is one of the promising therapeutic targets [40,41,42]. Of note, there are several reports on the relationship between VDAC1 and the PI3K/AKT/mTOR pathway, e.g., negative regulation of VDAC1 by the PI3K/Akt pathway via GSK3β and the positive feedback loop of VDAC1–AKT–GSK3β–VDAC1 [43,44]. Although further studies are needed to determine how statins regulate VDAC1, the effectiveness of statins against ovarian clear cell carcinoma is promising. *LDLRAP1*, which was extracted as a predictive marker for statin response with an inverse correlation to *VDAC1*, is involved in the incidence of familial hypercholesterolemia [45], although its association with cancer has not been reported. However, in relation to the response to statins, patients with autosomal recessive hypercholesterolemia, a rare disorder caused by *LDLRAP1* mutation, have been reported to respond better to statins than patients with homozygous hypercholesterolemia [46]. Although the mechanisms by which *VDAC1* and *LDLRAP1* are involved in statin response remain to be elucidated, clinical trials in which the expression of these genes narrows down the cases of ovarian cancer will be very promising.

In the present study, the safety of statins at the animal level was verified, and the response of statin to clinical specimens was confirmed; hence, statins are ready for clinical application. If statins are to be applied clinically to ovarian cancer, it may be considered not only as a single agent but also in combination with other drugs. Here, we searched for agents that produced synergistic effects with statins, and we found that paclitaxel and panobinostat were synergistic. Paclitaxel is an effective anticancer drug that binds stoichiometrically and specifically to the β-tubulin subunit in microtubules, and the synergistic effect of statins and paclitaxel may be related to *VDAC1* expression. VDAC could be regulated by tubulin via a functional interaction between dimeric tubulin and VDAC [47]. The interaction between tubulin and VDAC occurs at the C-terminal tail of tubulin, resulting in a negatively charged C-terminal, and tubulin interacts with the positively charged domain of VDAC [36]. Antitumor compounds targeting VDACs are beginning to emerge, and VDAC–tubulin antagonists could become a new generation of metabolism-oriented cancer chemotherapy drugs [48,49], with statins likely to be one of the candidates. Panobinostat, an HDAC inhibitor, is considered one of the most promising drugs for cancer treatment because its HDAC inhibitors induce cancer cell cycle arrest, cause differentiation and cell death, reduce angiogenesis, and regulate the DNA-damage response [50]. More than 200 clinical trials have already employed HDAC inhibitors, including some phase I trials. However, specific studies on gynecologic oncology are lacking, and panobinostat has been approved only for multiple myeloma. Panobinostat is an oral medication as well as a statin, and the combination of the two may have advantages in terms of compliance with treatment in cases intended for maintenance therapy or to prolong life in terminal conditions. It is hoped that the combination of panobinostat and statins will be tested in future clinical trials involving statins.

## 4. Materials and Methods

### 4.1. Drugs

Simvastatin (S6196) was purchased from Sigma-Aldrich (St. Louis, MO, USA), and lovastatin was purchased from LKT Laboratories, Inc. (St. Paul, MN, USA). Simvastatin and lovastatin were suspended in dimethyl sulfoxide (DMSO) (Merck Millipore, Waltham, MA, USA) and used at a concentration of 0–100 μM for the determination of IC_50_. L-778123 was kindly provided by Dr. Minoru Yasuda (Dai Nippon Printing Co., Ltd., Tokyo, JAPAN) who synthesized it in accordance with a previous report [18].

### 4.2. Cell Lines and Cell Cultures

A2780, OVCAR5, and SKOV3, which are widely used as human ovarian cancer cell lines, were used in the experiments. In addition, serous carcinoma (KURAMOCHI and OVSAHO), clear cell carcinoma (ES-2, OVISE, RMG-1, and TOV-21G), mucinous carcinoma (JHOM1, MCAS, and RUMG-S) were used as the representative cell lines of common histological types of ovarian cancer. The human ovarian cancer cell lines KURAMOCHI, MCAS, OVISE, OVSAHO, RMG-1, and RMUG-S were purchased from the Japanese Collection of Research Bioresources Cell Bank; ES-2, OVCAR5, SKOV3, and TOV-21G were purchased from the American Type Culture Collection; and JHOM1 was purchased from the Cell Engineering Division, RIKEN BioResource Research Center. A2780 was kindly provided by Drs. A. Okamoto, S.B. Howell, and E. Reed. All cells were cultured in RPMI-1640 (FUJIFILM Wako Pure Chemical Corporation, Tokyo, Japan) with 10% fetal bovine serum (FBS) (Gibco, Waltham, MA, USA) and 100 U/mL penicillin–streptomycin (Gibco) and maintained at 37 °C with 5% CO_2_. To avoid cross-contamination and incorrect authentication, all cell lines were used for experiments within 4 years of purchase. Ascites-derived ovarian cancer cell lines (NOVC series) were established according to our previous report [51]. All NOVC−cells were cultured in RPMI-1640 (Wako) with 10% FBS (Gibco) and 100 U/mL of penicillin–streptomycin (Gibco), maintained at 37 °C with 5% CO_2_.

### 4.3. Cell Viability Assays

The end-point detection of cell viability was performed using CellTiter-Glo 2.0 assays (Promega, WI, USA). Luminescence was measured using Synergy H1 microplate reader (BioTek, Winooski, VT, USA). Concentrations that produced 50% inhibition of growth (IC_50_) were calculated using Prism 8 (GraphPad Software, La Jolla, CA, USA).

### 4.4. Microarray Analysis of KURAMOCHI and OVSAHO Cells Incubated with Simvastatin or L-778123

Total RNA was extracted from KURAMOCHI and OVSAHO cells incubated with simvastatin or L-778123 using RNeasy Mini Kits (Qiagen, Hilden, Germany) in combination with RNase-free DNase (Qiagen). RNA quality was assessed using Agilent 2100 Bioanalyzer (Agilent, Santa Clara, CA, USA). RNA samples were labeled using Low Input Quick Amp Labeling kits (one-color; Agilent) and then hybridized to Agilent SurePrint G3 Human Gene Expression version 3.0 Microarray (Agilent) as per the manufacturer’s instructions. The arrays were scanned using Agilent DNA Microarray Scanner G2505C (Agilent). The scanned images were extracted using Feature Extraction software 11.5.1.1 (Agilent Technologies). The processed data were imported into GeneSpring GX 14.9 (Agilent Technologies) for log2 transformation and summarization. The signal cutoff was <0.01, and signals were normalized to the 75th percentile of signal intensity to standardize each chip for comparisons. Two categories (cell and siRNA) were defined for two-way analysis of variance of the gene expression data. To further filter statistically significant genes, Tukey’s multiple comparison post hoc tests and Benjamini–Hochberg false discovery rate correction (corrected *p* < 0.05) were performed. Genes exhibiting a fold change of at least 2.0 were identified as differentially expressed genes (DEGs). For enrichment and gene set analysis, the normalized data and DEGs were imported into MOGERA-Array Viewer 1.21.1 (TOHOKU CHEMICAL Co., Ltd.). Using this viewer, we performed GO analysis. A *p*-value of <0.05 was considered to indicate statistical significance.

### 4.5. Analysis of Cell Apoptosis and Autophagy

To determine cell apoptosis, OVCAR5 and SKOV3 cells were plated in 96-well plates at a density of 5 × 10^4^ cells/mL for 48 h, as described above, with 50 μM lovastatin and 100 μM. Caspase-Glo 3/7 Reagent (Promega, Madison, WI, USA) was added to each well. The cells were gently mixed using a plate shaker and incubated at room temperature. Finally, the luminescence of each sample was measured using a plate-reading luminometer. To analyze autophagy, OVCAR5 and SKOV3 cells stably expressing DsRed-LC3-GFP were subjected to lovastatin treatment at a density of 5 × 10^4^ cells/mL. The number of DsRed-LC3B puncta per cell was measured for each experimental group. At least 30 cells were measured. The values shown represent the mean and standard deviation (SD) from the examined cells.

### 4.6. Correlation Analysis between mRNA Expression and Cell Viability Using Pearson’s Product-Moment Correlation Coefficient

Total RNA extracted from A2780, ES-2, JHOM1, MCAS, OVISE, OVSAHO, RMG-1, RUMG-S, TOV-21G, NOVC-1C, 2C, 4C, 5C, 7C, and 8C cells was analyzed using GeneChip Human Genome U133 Plus 2.0 arrays (Affymetrix, Santa Clara, CA, USA). These chips include 54,675 probe sets for the analysis of the mRNA expression levels of approximately 47,000 transcripts and variants from 38,500 well-characterized human genes. Target hybridization, washing, and staining with signal amplification were performed according to the manufacturer’s protocols. The arrays were scanned using GeneChip Scanner 3000 (Affymetrix), and the intensity of each feature of the array was calculated using GeneChip Analysis Suite version 4.0 software (Affymetrix). The mean expression value in each experiment was normalized to 1000 to reliably compare multiple arrays. Entities in which 90% of the samples had values of <500 were excluded from analyses. The correlation between these mRNA expression values and the cell viability of each cell line at 72 h in simvastatin (10 μM)-treated culture was analyzed using Pearson’s product–moment correlation coefficient analysis.

### 4.7. Quantitative RT-PCR

Total RNA was isolated from the cultured cells using ISO-GEN (Nippon Gene, Tokyo, Japan), followed by isopropanol precipitation-based purification. cDNA was synthesized using SuperScript IV VILO Master Mix (Thermo Fisher Scientific, Waltham, MA, USA) as per the manufacturer’s instructions. Real-time PCR assays were performed using iQ SYBR Green Supermix (Bio-Rad Laboratories, Hercules, CA, USA) on CFX96 Real-Time PCR Detection System (Bio-Rad Laboratories). The primer sequences were as follows: *ACTB* forward 5′-CACCATTGGCAATGAGCGGTTC-3′ and reverse 5′-AGGTCTTTGCGGATGTCCACGT-3′; *VDAC1* forward 5′-GCAAAATCCCGAGTGACCCAGA-3′ and reverse, 5′-TCCAGGCAAGATTGACAGCGGT-3′; and *LDLRAP1* forward 5′-TATCCTGACAGACAACCTCACC-3′ and reverse 5′-CGATGTATGCAAACACCTTGTC-3′. Relative mRNA levels were calculated by the normalization of the cycle threshold (Ct) values of the target genes (*VDAC1* and *LDLRAP1*) to those of the reference gene (*ACTB*). Data are presented as the mean ± SD of triplicate measurements.

### 4.8. Histoculture Drug Response Assays

Ovarian cancer specimens were submitted to SRL, Inc. (Tokyo, Japan), and the histoculture drug response assays for simvastatin and the comparators paclitaxel and carboplatin were performed as per the company’s protocol. In brief, tumor tissues harvested intraoperatively were washed thoroughly with Hank’s Balanced Salt Solution, and necrotic and normal tissue portions were quickly removed. The tumor tissues were cut into small pieces, and approximately 10–20 mg of tissue was placed on collagen gel in a 24-well plate. Each test drug was dissolved in RPMI-1640 medium supplemented with 20% FBS. Each solution (1 mL) was placed in separate wells, and the plates were incubated at 37°C with 5% CO_2_ for 7 days. After incubation, collagenase and MTT solutions were added, and the plates were incubated for 16 h. The medium was then removed, DMSO was added, and MTT–formazan produced by the enzymatic reaction in the cells was extracted. The absorbance of each well was measured using a microplate reader at 540 nm with 630 nm as a control. The efficacy of each drug was calculated according to the inhibition index as follows: inhibition index (%) = (1 − mean absorbance per gram of treated tumor/mean absorbance per gram of control tumor) × 100. The concentration of the drugs used in this assay was as follows: simvastatin, 100 μM/mL; paclitaxel, 40 μg/mL; and carboplatin, 30 μg/mL. Written informed consent was obtained from each patient regarding the use of samples for research. The Ethics Committee of Keio University approved this study (approval no. 20070081).

### 4.9. Combination Index Analysis

SynergyFinder™ was used to evaluate the synergistic effects of simvastatin and the anticancer drugs AZD8055, copanlisib, dabrafenib, doxorubicin, etoposide, irinotecan, niraparib, paclitaxel, panobinostat, and trametinib in ES-2 cells at Netherlands Translational Research Center B.V. (Kloosterstraat 9, Oss, The Netherlands), and the process was performed as per the manufacturer’s protocol. The inhibition of cell proliferation is shown in the assay readout. For each combination, the dose–response curves of compounds as single agents and in mixtures with other compounds were measured in three fixed-ratio combinations. Synergy was determined using the combination index (CI) [52]. CI is defined for a certain percentage cell viability (V), which is the signal related to a non-exposed control: V = 100% × luminescence_treated,t=end_/luminescence_untreated,t=end_. The concentrations of the two compounds cpd1 and cpd2 needed to reach V in combination are then compared to the concentrations needed as single agents: CI_(100-V)_ = [cpd1]_V_/IC_(100-V),cpd1_ + [cpd2]_V_/IC_(100-V),cpd2_. CI was represented by the average CI at different effect levels, i.e., CI_50_, CI_75_, and CI_90_. CI = 1.0: no synergy; CI < 1.0: synergy; CI < 0.3: strong synergy; and CI > 1.5: antagonistic.

### 4.10. Hemogram and Biochemical Analysis

Studies on animals from which blood was collected have been reported previously [14]. In brief, mice were euthanized at the end of the study. Blood samples were collected by intracardiac aspiration using a 1-mL syringe with a 25-gauge needle and placed in a microcentrifuge tube containing EDTA. Blood was centrifuged, and hemogram and biochemical parameters were measured using standard clinical laboratory assays on Roche Hitachi Cobas c701 analyzer (Roche Diagnostics, Basel, Switzerland). All animal care and procedures were performed in accordance with institutional guidelines.

### 4.11. Statistical Analyses

Statistical analyses were performed using SPSS statistical software version 27.0 (IBM Japan, Tokyo, Japan) and Prism 8.0 GraphPad software. The Mann–Whitney U test was performed to assess the apoptosis and autophagy assay and to validate the expression of *VDAC1* and *LDLRAP1*. Specific analyses performed for each assessment are described in the specified methods section. In all analyses, data were evaluated using a two-tailed test; *p* < 0.05 was considered statistically significant.

## 5. Conclusions

There is little doubt that statins have an inhibitory effect on cell growth and have antitumor activity in ovarian cancer, but we have not been able to translate these results into clinical practice. The assumption that statins are effective against ovarian cancer as a whole should be discontinued. Based on the results of our studies, it is expected that clinical trials will be conducted and validated with reference to the expression of clinical biomarkers such as *VDAC1* and *LDLRAP1*, which can predict the response to statins in histological types other than mucinous carcinoma, for the purpose of add-on effects to standard paclitaxel–carboplatin therapy, maintenance therapy after initial treatment, or prevention of disease in high-risk patients with ovarian cancer.

## Figures and Tables

**Figure 1 pharmaceuticals-15-00124-f001:**
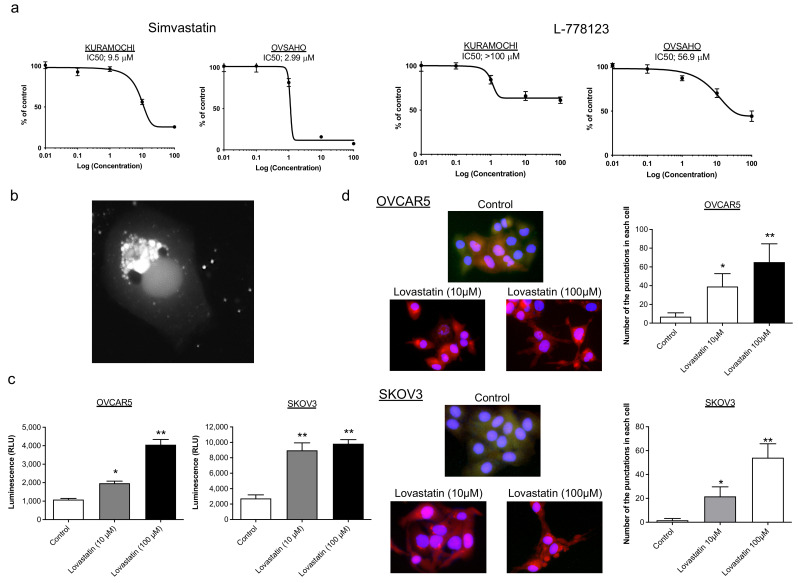
Exploration of new pathways and modes of action of statin. (**a**) IC50 after 72 h of incubation with simvastatin or L-778123 in KURAMOCHI and OVSAHO cells. (**b**) SKOV3 cells stably expressing DsRed-LC3-GFP imaged under fluorescence excitation after 48 h of treatment with lovastatin. Vacuolization and fragmentation are prominent. (**c**) Caspase-3/7 activity was monitored via luminescence activity after 48 h of treatment with lovastatin (*n* = 3). * *p* < 0.05, ** *p* < 0.01. (**d**) Representative imaging of OVCAR5 and SKOV3 merging the images of DAPI, DsRed-LC3, and GFP after 48 h of treatment with lovastatin. DAPI, nuclear staining; DsRed-LC3, red signals from DsRed-LC3B puncta; GFP, green signals from uncleaved DsRed-LC3B-GFP reporter. The number of DsRed-LC3B puncta per cell was measured for each experimental group (*n* = 3). * *p* < 0.05, ** *p* < 0.01.

**Figure 2 pharmaceuticals-15-00124-f002:**
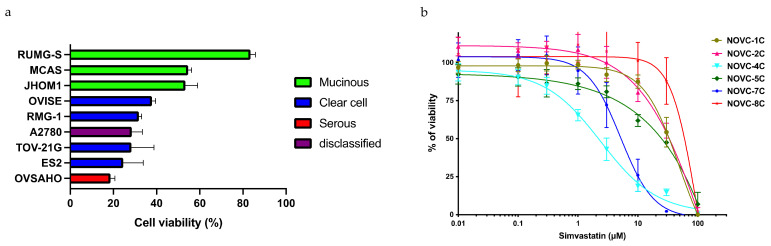
Response to simvastatin in existing cell lines and human ovarian cancer ascites-derived cell lines. (**a**) Cell viability of nine existing ovarian cancer cell lines (RUMG-S, MCAS, JHOM1, OVISE, RMG-1, A2780, TOV-21G, ES-2, and OVSAHO) after 72 h of simvastatin (10 μM)-mediated culture (*n* = 3). The cell lines are color coded as per histopathological type: red for serous carcinoma, blue for clear cell carcinoma, green for mucinous carcinoma, and purple for disclassified. (**b**) Cell viability of cell lines established from human ovarian cancer ascites (NOVC-1C, 2C, 4C, 5C, 7C, and 8C) after 72 h of simvastatin-mediated culture.

**Figure 3 pharmaceuticals-15-00124-f003:**
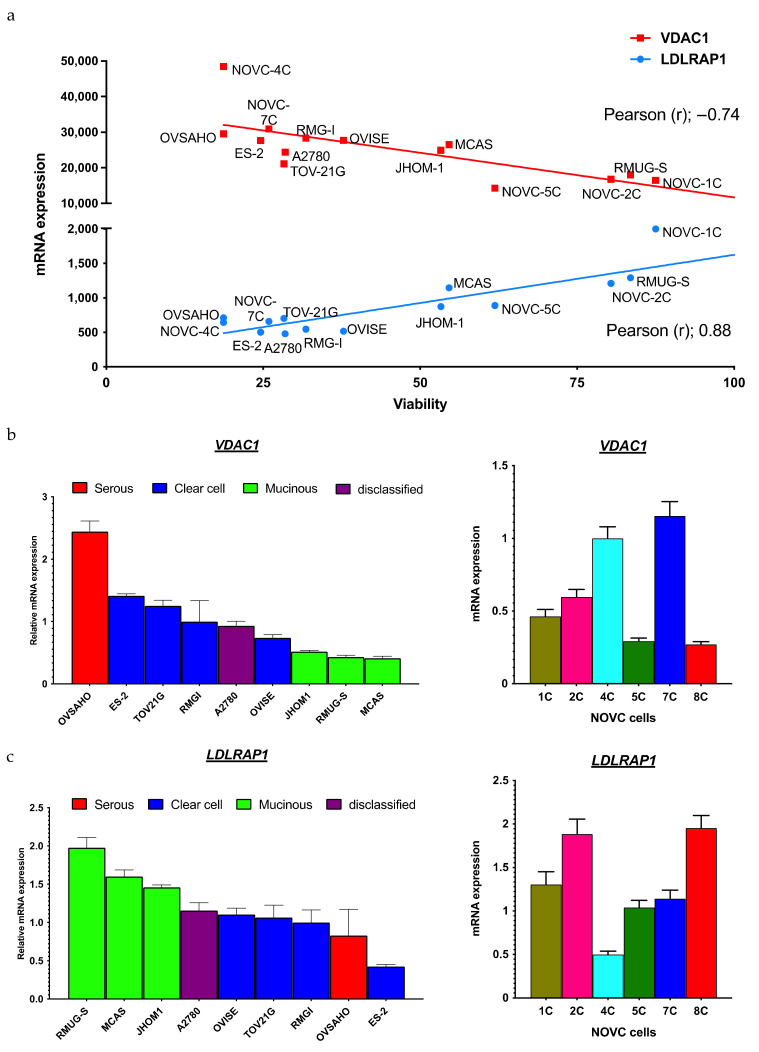
Establishment of a biomarker to predict response to statins in ovarian cancer. (**a**) *VDAC1* and *LDLRAP1* were identified from existing ovarian cancer cell lines and human ovarian cancer ascites-derived cell lines using microarray analysis. mRNA expressions were calculated by normalization of fluorescence intensity of *VDAC1* and *LDLRAP1* to those of the reference gene *ACTB*. Pearson’s correlation coefficient (r) was calculated from gene expression and cell viability following simvastatin administration. *VDAC1* was positively correlated with simvastatin response, and *LDLRAP1* was negatively correlated with simvastatin response. (**b**) Validation of *VDAC1* expression in nine existing ovarian cancer cell lines (A2780, ES-2, JHOM1, MCAS, OVISE, OVSAHO, RMG-1, RUMG-S, and TOV-21G) and human ovarian cancer ascites-derived cell lines (NOVC-1C, 2C, 4C, 5C, 7C, and 8C). The mRNA expression in each cell was normalized to *ACTB* expression (*n* = 3). (**c**) Validation of *LDLRAP1* expression in nine existing ovarian cancer cell lines (A2780, ES-2, JHOM1, MCAS, OVISE, OVSAHO, RMG-1, RUMG-S, and TOV-21G) and the cell lines established from human ovarian cancer ascites (NOVC-1C, 2C, 4C, 5C, 7C, and 8C). The mRNA expression in each cell was normalized to *ACTB* expression (*n* = 3).

**Figure 4 pharmaceuticals-15-00124-f004:**
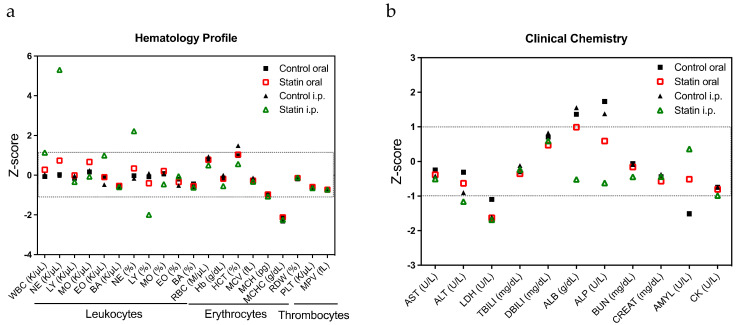
Hematological toxicity in mogp-Tag mice treated with lovastatin. (**a**) Hematology profile. (**b**) Clinical chemistry.

**Figure 5 pharmaceuticals-15-00124-f005:**
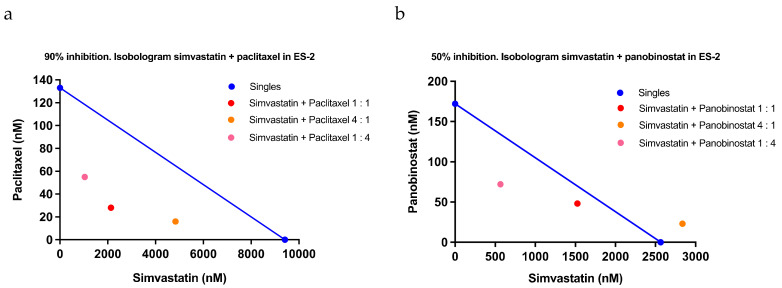
Synergistic effects of simvastatin and paclitaxel and panobinostat in ES-2 cells. An isobologram is a dose-oriented plot which reveals whether drug combinations are synergistic. On the axis, the calculated doses of the single compounds were plotted as the blue points that give the pre-set growth effect. Both points were connected with a straight blue line as an additivity line. For the drug combinations, it was calculated which dilutions gave the pre-set growth effect and the concentrations of the individual components at this point were plotted in the isobologram. The combinations of simvastatin and each drug by condition in the legend were plotted with the red, orange, and pink points. In case of an additive drug effect, the drug combination will lie close to the additivity line. In case of synergy or antagonism, the points will lie under or above the line, respectively. (**a**) Isobologram for 90% growth inhibition by simvastatin and paclitaxel. (**b**) Isobologram for 50% growth inhibition by simvastatin and panobinostat.

**Table 1 pharmaceuticals-15-00124-t001:** Candidate genes that predict the response to simvastatin according to Pearson’s product-moment correlation coefficient analysis.

Gene	Pearson’s Correlation Coefficient (r)	*p*-Value	Spearman’s Correlation Coefficient (r)	*p*-Value
Top 10				
*NRDC*	−0.90	7.38 × 10^−7^	−0.78	2.49 × 10^−4^
*PPID*	−0.79	1.67 × 10^−7^	−0.82	6.33 × 10^−5^
*XRN2*	−0.78	2.30 × 10^−4^	−0.76	4.25 × 10^−4^
*VDAC1*	−0.74	6.44 × 10^−4^	−0.75	5.73 × 10^−4^
*HSP90AB1*	−0.74	7.19 × 10^−4^	−0.77	3.10 × 10^−4^
*DOCK7*	−0.74	7.45 × 10^−4^	−0.67	3.28 × 10^−3^
*VARS1*	−0.73	9.71 × 10^−4^	−0.82	5.24 × 10^−5^
*HSDL1*	−0.72	9.92 × 10^−4^	−0.74	7.16 × 10^−4^
*ING2*	−0.72	1.02 × 10^−3^	−0.62	8.36 × 10^−3^
*AGK*	−0.72	1.04 × 10^−3^	−0.79	1.83 × 10^−4^
Bottom 10				
*LDLRAP1*	0.88	3.99 × 10^−6^	0.82	6.04 × 10^−5^
*EPN3*	0.87	4.52 × 10^−6^	0.73	7.85 × 10^−4^
*P4HTM*	0.82	5.09 × 10^−5^	0.68	2.57 × 10^−3^
*VPS37C*	0.80	1.23 × 10^−4^	0.66	3.96 × 10^−3^
*PHF2*	0.80	1.24 × 10^−4^	0.77	3.21 × 10^−4^
*JADE2*	0.80	1.33 × 10^−4^	0.71	1.37 × 10^−3^
*CXorf56*	0.80	1.34 × 10^−4^	0.67	2.98 × 10^−3^
*PPL*	0.79	1.52 × 10^−4^	0.83	3.91 × 10^−5^
*OVOL1*	0.79	1.60 × 10^−4^	0.75	5.73 × 10^−4^
*IFNLR1*	0.78	2.49 × 10^−4^	0.75	5.02 × 10^−4^
*SLC1A4*	0.77	2.67 × 10^−4^	0.77	2.99 × 10^−4^

**Table 2 pharmaceuticals-15-00124-t002:** Clinical characteristics and inhibition index based on a histoculture drug response assay.

No	Histological Type	Age (Years)	Stage	Simvastatin (%)	Paclitaxel (%)	Carboplatin
1	High-grade serous	65	IIB	77.6	82.7	61.0
2		45	IIIA2	65.1	72.4	60.3
3		46	IVB	53.5	65.0	15.5
4		80	IVA	44.6	86.3	58.9
5		71	IIIC	38.1	41.5	33.2
6		40	IIIB	24.9	69.2	52.7
7		71	IIIC	12.6	54.1	11.2
8		48	IIIA	10.3	81.1	18.4
9		74	IIIC	6.0	51.4	28.6
10	Clear cell	54	IA	67.6	82.2	19.1
11		74	IC2	67.5	76.3	29.4
12		49	IC3	61.4	77.6	32.7
13		51	IC1	61.0	34.2	56.2
14		59	IIIC	42.5	50.5	59.1
15		46	IVB	18.0	41.7	33.3
16	Endometrioid	33	IC1	77.7	83.2	62.9
17		79	IC1	72.9	78.4	45.2
18		63	IC1	72.5	85.2	53.3
19		59	IIB	15.7	51.0	29.4
20	Mucinous	52	IC2	22.7	85.3	38.8

**Table 3 pharmaceuticals-15-00124-t003:** Combination index (CI) between simvastatin and examined compounds calculated from the mixture data. CI_50_ corresponds to 50% viability, CI_75_ to 25% viability, and CI_90_ to 10% viability. A representative value is the average CI value for the three mixtures.

Combination Experiment	CI_50_	CI_75_	CI_90_	Average	Standard Deviation
AZD8055					
Simvastatin + AZD8055 1:1	0.78	0.77	0.98	0.84	0.12
Simvastatin + AZD8055 4:1	1.36	1.12	0.91	1.13	0.23
Simvastatin + AZD8055 1:4	0.72	0.73	>2.00	>1.15	NA
Average	0.95	0.87	>1.30		
Standard deviation	0.35	0.22	NA		
Copanlisib					
Simvastatin + Copanlisib 1:1	1.04	0.77	>2.00	>0.27	NA
Simvastatin + Copanlisib 4:1	1.04	1.17	1.52	1.24	0.24
Simvastatin + Copanlisib 1:4	0.90	0.60	>2.00	>1.17	NA
Average	0.99	0.85	>1.84		
Standard deviation	0.08	0.29	NA		
Dabrafenib					
Simvastatin + Dabrafenib 1:1	1.04	0.67	0.74	0.82	0.20
Simvastatin + Dabrafenib 4:1	0.71	0.48	>2.00	>1.06	NA
Simvastatin + Dabrafenib 1:4	0.66	>2.00	>2.00	>1.55	NA
Average	0.80	> 1.05	>1.58		
Standard deviation	0.21	NA	NA		
Doxorubicin					
Simvastatin + Doxorubicin 1:1	1.05	1.14	1.11	1.10	0.04
Simvastatin + Doxorubicin 4:1	1.73	1.47	1.08	1.43	0.33
Simvastatin + Doxorubicin 1:4	1.04	1.01	0.96	1.00	0.04
Average	1.27	1.20	1.05		
Standard deviation	0.40	0.24	0.08		
Etoposide					
Simvastatin + Etoposide 1:1	1.44	1.11	0.78	1.11	0.33
Simvastatin + Etoposide 4:1	1.16	1.14	0.98	1.09	0.10
Simvastatin + Etoposide 1:4	0.98	0.95	0.91	0.95	0.04
Average	1.19	1.07	0.89		
Standard deviation	0.23	0.10	0.10		
Irinotecan					
Simvastatin + Irinotecan 1:1	1.48	1.11	0.79	1.13	0.34
Simvastatin + Irinotecan 4:1	1.13	0.96	0.79	0.96	0.17
Simvastatin + Irinotecan 1:4	1.20	1.06	0.95	1.07	0.13
Average	1.27	1.04	0.84		
Standard deviation	0.18	0.08	0.09		
Niraparib					
Simvastatin + Niraparib 1:1	1.30	1.18	0.86	1.11	0.23
Simvastatin + Niraparib 4:1	0.93	0.81	0.70	0.81	0.11
Simvastatin + Niraparib 1:4	1.07	1.06	0.83	0.99	0.13
Average	1.10	1.02	0.80		
Standard deviation	0.19	0.19	0.08		
Paclitaxel					
Simvastatin + Paclitaxel 1:1	0.62	1.13	0.52	0.76	0.33
Simvastatin + Paclitaxel 4:1	1.01	1.35	0.14	0.83	0.62
Simvastatin + Paclitaxel 1:4	0.73	0.67	0.37	0.59	0.19
Average	0.79	1.05	0.34		
Standard deviation	0.20	0.34	0.19		
Panobinostat					
Simvastatin + Panobinostat 1:1	0.88	0.56	0.30	0.58	0.29
Simvastatin + Panobinostat 4:1	1.24	0.82	0.34	0.80	0.45
Simvastatin + Panobinostat 1:4	0.64	0.32	>2.00	>0.98	NA
Average	0.92	0.56	>0.88		
Standard deviation	0.30	0.25	NA		
Trametinib					
Simvastatin + Trametinib 1:1	1.34	1.31	1.27	1.31	0.03
Simvastatin + Trametinib 4:1	0.97	0.84	0.70	0.84	0.14
Simvastatin + Trametinib 1:4	1.02	1.15	1.36	1.17	0.17
Average	1.11	1.10	1.11		
Standard deviation	0.20	0.24	0.36		

## Data Availability

Data are contained within the article and Appendix A.

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
