# Peer review of "Response Predictive Markers and Synergistic Agents for Drug Repositioning of Statins in Ovarian Cancer"

_pharmaceuticals, 2022, doi:10.3390/ph15020124_

Round 1

Reviewer 1 Report

The aim of this study was to establish biomarkers for identification of ovarian cancer patients who will respond to statin in clinical regimen. Authors found two genes VDAC1 and LDLRAP1 correlating with response to statin. In addition, they found synergistic effects of statin with paclitaxel and panobinostat promissing for future clinical testing. Findings of this study are very interesting, but they should be described and discussed more preciously. This manuscript is capable of being published after the major revision process mainly in discussion part of the manuscript.

Major points:

1) Methodological part;

Paragraph 4.2. Authors should describe in detail type of used cell lines and include information, why exactly these particular cell lines were selected for this study.

Paragraph 4.7: There is not declared the method of gene expression quantification.

2) Results;

Line 206: Please specify the reason, why cell line ES-2 was selected for this type of the study.

The combination experiments are described very briefly and they should be described in more detail. Every combination should be discussed and evaluated in the results section.

3) Discussion;

This part is focused mainly on VDAC1 gene. There is not mentioned the next significant gene LDLRAP1 and its function and the effect on statin efficiency is not discussed. Authors should discuss those results of their study. Furthermore, findings of the section 2.3. especially combination effects are summarized and discussed too briefly and this part of the discussion should be enlarged and all potential advantages of statin in combination with paclitaxel or panobinostat should be described.

Minor points:

1) Authors should controll a few formal mistakes e.g.: line 66: protein instead of protien, line 214: hitoculture instead of histocluture, line 345 and 350: 5x104cells/mL instead of 5x104cells/mL

2) Please, specify the units of CI50,75,90 values in Table 5.

Author Response

Response to Reviewer #1

Major points:

1) Methodological part;

Paragraph 4.2. Authors should describe in detail type of used cell lines and include information, why exactly these particular cell lines were selected for this study.

              We appreciate the reviewer’s comment. A2780, OVCAR5, and SKOV3, which are widely used as human ovarian cancer cell lines, were used in the experiments. In addition, serous carcinoma; KURAMOCHI, OVSAHO, clear cell carcinoma; ES-2, OVISE, RMG-1, TOV-21G, mucinous carcinoma; JHOM1, MCAS, RUMG-S were used as representative cell lines of common histological types of ovarian cancer. This description is added to the revised manuscript (line 341-345).

Paragraph 4.7: There is not declared the method of gene expression quantification.

              We appreciate the reviewer’s comment. The method of gene expression quantification has been added to the revised manuscript (line 422-425).

2) Results;

Line 206: Please specify the reason, why cell line ES-2 was selected for this type of the study.

              We appreciate the reviewer’s comment. We reported that statin may suppress glutathione and avoid the Warburg effect, while clear cell carcinoma is resistant to anticancer drugs by enhancing the Warburg effect using glutathione. Thus, in this study, we evaluated the combination index of each drug in ES-2, which had the highest response to statin among the clear cell carcinoma cell lines. This description is added to the revised manuscript (line 208-212).

The combination experiments are described very briefly and they should be described in more detail. Every combination should be discussed and evaluated in the results section.

              We appreciate the reviewer’s constructive comment. All combination indexes for simvastatin and each drug were added to the revised manuscript (line 212-221).

3) Discussion;

This part is focused mainly on VDAC1 gene. There is not mentioned the next significant gene LDLRAP1 and its function and the effect on statin efficiency is not discussed. Authors should discuss those results of their study. Furthermore, findings of the section 2.3. especially combination effects are summarized and discussed too briefly and this part of the discussion should be enlarged and all potential advantages of statin in combination with paclitaxel or panobinostat should be described.

              We appreciate the reviewer’s constructive comment. There are no reports on the relationship between LDLRAP1 and cancer, but based on the association with statin response, we are thinking that ovarian cancers with low expression of LDLRAP1 are more likely to respond to statins, and the cell growth inhibitory effect of statins may be stronger in these patients. This description was added to the revised manuscript (line 299-307). In addition, in the discussion of the results of section 2.3, we specifically mentioned combination therapy, incorporating the following; the synergistic effect of statin and paclitaxel may be related to VDAC1 expression. Panobinostat is an oral medication as well as a statin, and the combination of the two may have advantages in terms of compliance with treatment in cases intended for maintenance therapy or to prolong life in terminal conditions. We have added to the discussion from these perspectives (line 308-331).

Minor points:

1) Authors should controll a few formal mistakes e.g.: line 66: protein instead of protien, line 214: hitoculture instead of histocluture, line 345 and 350: 5x104cells/mL instead of 5x104cells/mL

              We appreciate the reviewer’s comment and have corrected the mistakes in the revised manuscript.

2) Please, specify the units of CI50,75,90 values in Table 5.

We appreciate the reviewer’s comment. CI is defined for a certain percentage cell viability (V), which is the signal related to a non-exposed control: V = 100 % x luminescencetreated,t=end / luminescenceuntreated,t=end. The concentrations of the two compounds cpd1 and cpd2 needed to reach V in combination are then compared to the concentrations needed as single agents:

CI (100-V) = [cpd1]V / IC(100-V),cpd1 + [cpd2]V / IC(100-V),cpd2. Hence, there is no unit in CI.

Reviewer 2 Report

manuscript can be published

manuscript is well organized and presented

In Discussion elaborate  more on the signaling network of the  PI3K  and  AKT and their  correlation with the VDAC1 and LDLRAP1 genes

Author Response

Response to Reviewer #2

manuscript can be published, manuscript is well organized and presented

In Discussion elaborate more on the signaling network of the PI3K and AKT and their correlation with the VDAC1 and LDLRAP1 genes.

We appreciate the reviewer for the constructive comment. We have added an argument to the discussion that statin can be expected to have a response in clear cell carcinoma, since the PI3K/AKT/mTOR pathway, which is often altered in clear cell carcinoma, has been reported to be associated with the VDAC1 gene (line 291-299). The reviewer's suggestion made the discussion more lively, thank you very much.

Reviewer 3 Report

The manuscript by Kobayashi et al. aims to suggest possible biomarkers for the correct repurposing of statin for the treatment of ovarian cancer. By gene expression analysis of statin treated versus untreated cells they identified genes positively and negatively correlated to treatment response and suggest VDAC1 as possible predictive biomarker.

Major concerns

Although the manuscript includes different molecular analyses, it results to be only descriptive without any mechanistic data.

It is unclear how the data obtained from the gene expression analysis whose results are described in Tables 1 and 2 are then used.

Although with different assays, the ability of statin in inducing apoptosis and autophagy in SKOV3 and OVCAR5 (described in figure 1 b-d) was already shown and described by the authors in their CCR 2015 paper, therefore this part of the manuscript cannot be considered new.

Pathological characteristics of the NOVC cell lines obtained from ovarian cancer patients’ ascites is not reported (figure 2b). is their sensitivity to statin correlated to their histology subtype?

Correlation of VDAC1 and LDLRAP1 expression with sensitivity to statin was done using gene expression data of long-  and short-term cell cultures, while RT-PCR validation was performed only on patient-derived cell lines; the same validation should be done also on long term established cell lines (Figure 3)

To demonstrate the direct involvement of these two genes in response to statin their expression should be modulated by silencing or over-expression in the appropriated cellular models. 

Considering that in this manuscript are presented only data obtained on cell lines or on clinical specimens, data included in figure 4 are poorly relevant.

Data reported in table 5 are very interesting, however considering the effort in searching for biomarkers of prediction one would expect a molecular characterization of these samples.

Minor comments

At page 2 reference numbering is not sequential.

Legend of horizontal axes in hematology profile of figure 4a contains Japanese characters.

Author Response

Response to Reviewer #3

Major concerns

Although the manuscript includes different molecular analyses, it results to be only descriptive without any mechanistic data.

We appreciate the reviewer’s comment. We received many positive and constructive suggestions from the five reviewers, and based on them, we have revised the structure and description of the manuscript to make it more scientific. Thank you very much again for your kind help.

It is unclear how the data obtained from the gene expression analysis whose results are described in Tables 1 and 2 are then used.

We agree with the reviewer’s comment. Following a kind suggestion from other reviewers, we have moved tables 1 and 2 into the supplementary material in the revised manuscript.

Although with different assays, the ability of statin in inducing apoptosis and autophagy in SKOV3 and OVCAR5 (described in figure 1 b-d) was already shown and described by the authors in their CCR 2015 paper, therefore this part of the manuscript cannot be considered new.

We appreciate the reviewer’s comment. Our previous report in Clinical Cancer Research was a western blot assay and this time we applied a luciferase assay and DsRed-LC3B puncta to measure the autophagosome. We think this is worth reporting because of the difference in what we are assessing.

Pathological characteristics of the NOVC cell lines obtained from ovarian cancer patients’ ascites is not reported (figure 2b). is their sensitivity to statin correlated to their histology subtype?

We appreciate the reviewer’s comment. As described in the Cancer Cell article (Ref 51), cell lines established from human ovarian cancer ascites (NOVC series) were from ovarian cancer patients who underwent cell-free and concentrated ascites reinfusion therapy at the Kaname-cho Hospital (Tokyo, Japan) and were cultured in vitro at the National Cancer Center Hospital. During this process, the histopathological type of some patients was missing due to insufficient information provided by the referring hospital. Thus, in this study, the NOVC series were used only for molecular biological evaluation and existing cell lines were used for correlation with histopathological type.

Correlation of VDAC1 and LDLRAP1 expression with sensitivity to statin was done using gene expression data of long- and short-term cell cultures, while RT-PCR validation was performed only on patient-derived cell lines; the same validation should be done also on long term established cell lines (Figure 3)

We appreciate and agree with the reviewer’s comment. According to the reviewer’s suggestion, the expression of VDAC1 and LDLRAP1 in existing cell lines was evaluated by qRT-PCR and added the data to Figure 3b and Figure 3c of the revised manuscript.

To demonstrate the direct involvement of these two genes in response to statin their expression should be modulated by silencing or over-expression in the appropriated cellular models.

We appreciate the reviewer’s constructive comment. However, please understand that we are not able to do much experiment because the editor asked us to respond to the comments from 5 reviewers within 10 days and resubmit it. We feel the experiment you proposed is important, so we will incorporate it into our future experimental schedule and report back in the future.

Considering that in this manuscript are presented only data obtained on cell lines or on clinical specimens, data included in figure 4 are poorly relevant.

We would like to thank the reviewer for this comment. Our aim is to apply the results of our work to date, represented by this manuscript, to the implementation of the clinical trial that we have already prepared. The safety of statin administration is a very important aspect for future clinical applications. We are pleased that we finally complete assessing the safety of statin using blood and biochemical tests in this manuscript, as we assessed safety only using mouse behavior and body weight in our previous Clinical Cancer Research paper.

Data reported in table 5 are very interesting, however considering the effort in searching for biomarkers of prediction one would expect a molecular characterization of these samples.

We appreciate the reviewer’s comment. We have actually performed the multigene testing for some cases shown in table 5 and are getting the results of the analysis, but it will take much time to complete the data collection for the recent cases. Thank you for your constructive suggestions, and we will report back when we have all the data.

Minor comments

At page 2 reference numbering is not sequential.

We would like to thank the reviewer for pointing out the errors. The reference numbers have been reassigned in the revised manuscript.

Legend of horizontal axes in hematology profile of figure 4a contains Japanese characters.

We would like to thank the reviewer for pointing out the errors. The figure 4a has been corrected in the revised manuscript.

Reviewer 4 Report

The authors studied the influence of statin in ovarian cancer by investigating the mode of action, response prediction biomarker, response in clinical sample and synergistic agent. This work may be helpful to identify patients who may respond to statin treatment in clinical practice by measuring the biomarkers like VDAC1 as suggested in this work. However, generally speaking, this manuscript is kind of hard to read, which is mainly because the experiments and results including the figures have not been described or explained very clearly. Following are some contents that authors can improve:

1. Line 76, the authors can not conclude that statin has other modes of action besides protein prenylation just based on the results that statin displayed higher cytotoxicity than L-778123, a simultaneous inhibitor of both farnesyltransferase and geranylgeranyltransferase. Because there might be other factors that can influence the cytotoxicity (e.g. the cellular uptake of the compounds)

2. Figure 1a, error bars should be added to the figures. Figure 1b did not give a clear picture of fragmentation and vacuoles.

3. More detailed explanation of Figure 4 and Figure 5 are needed, it is a bit hard to correlated the figures with the conclusions in the main text.

4. About 4.6 part (line 354), to study the relationship between mRNA expression of VDAC1 and LDLRAP1 and the cell viability in around 15 cell lines, the authors used GeneChip arrays which can analyze the mRNA expression levels of 359 approximately 47,000 transcripts, which does not seem to be reasonable. Not only because the expression of too much mRNA were analyzed with this array experiment considering only two mRNA that the authors were interested, but also because such experiments cost a lot. The authors may need to explain that! Besides, I suppose the result related to 4.6 part points to Figure 3a, the authors did not clearly explain the figure, such as the concentrations that has been applied, which spot correlates with which cell line?

5. Clinical samples and animals were used in this work, any information that indicates that all the performance accords with the standard of morality and ethic was not noticed in the manuscript.

Author Response

Response to Reviewer #4

  1. Line 76, the authors can not conclude that statin has other modes of action besides protein prenylation just based on the results that statin displayed higher cytotoxicity than L-778123, a simultaneous inhibitor of both farnesyltransferase and geranylgeranyltransferase. Because there might be other factors that can influence the cytotoxicity (e.g. the cellular uptake of the compounds)

We agree with the reviewer’s comment. We have weakened the nuance of the relevant sentence to the argument that it is possible (line 73-74).

  1. Figure 1a, error bars should be added to the figures. Figure 1b did not give a clear picture of fragmentation and vacuoles.

We appreciate the reviewer’s comment. Error bars in figure 1a have been added to the revised manuscript. Figure 1b has been replaced by a photo with a higher magnification.

  1. More detailed explanation of Figure 4 and Figure 5 are needed, it is a bit hard to correlated the figures with the conclusions in the main text.

We agree with the reviewer's comment. The explanation of Figure 4 is added in the text (line 197-202), and the interpretation of Figure 5 is added in the discussion (line 308-331).

  1. About 4.6 part (line 354), to study the relationship between mRNA expression of VDAC1 and LDLRAP1 and the cell viability in around 15 cell lines, the authors used GeneChip arrays which can analyze the mRNA expression levels of 359 approximately 47,000 transcripts, which does not seem to be reasonable. Not only because the expression of too much mRNA were analyzed with this array experiment considering only two mRNA that the authors were interested, but also because such experiments cost a lot. The authors may need to explain that! Besides, I suppose the result related to 4.6 part points to Figure 3a, the authors did not clearly explain the figure, such as the concentrations that has been applied, which spot correlates with which cell line?

We appreciate the reviewer’s kind and helpful comment. The GeneChip Arrays were certainly an expensive experiment, but we feel that it has become a matter of course to do this kind of analysis beforehand in recent basic research papers. For Materials and Methods 4.6, the concentration of simvastatin has been described (line 410), and for figure 3a, which spot represents which cell line has been clearly indicated in the revised manuscript. Thank you very much again for the constructive comment.

  1. Clinical samples and animals were used in this work, any information that indicates that all the performance accords with the standard of morality and ethic was not noticed in the manuscript.

We appreciate the reviewer’s comment. Descriptions of patient consent, ethics committee approval, and animal protection procedures have been added to the revised manuscript (line 443-445, 463-464).

Reviewer 5 Report

This manuscript is an interesting study on various aspects of Statin with regard to cancer and its response prediction. 

However, I have a few concerns that need to be addressed:

1- The title of the manuscript is too long and confusing. I suggest that the authors shorten the title and make it more informative. The details can be added in the abstract

2. The language of the manuscript needs to be polished since the authors have failed in certain cases to clearly present their intent. 

3. It is not clear why the authors have tried to tie all these information ranging from Statin mode of action to response to treatment, etc. This has made the manuscript to some extent compilation of data rather than a core-oriented work with a clear aim. This needs to be explained and the paper should be revised (especially the introduction) accordingly. 

4. Some tables can be removed and added to the supplementary material instead. 

Author Response

Response to Reviewer #5

1- The title of the manuscript is too long and confusing. I suggest that the authors shorten the title and make it more informative. The details can be added in the abstract

We would like to thank you for this comment as well as the third comment. In the revised manuscript, I clearly stated the purpose of this paper, and the title reflects it.

  1. The language of the manuscript needs to be polished since the authors have failed in certain cases to clearly present their intent.

We appreciate the reviewer's kind comment. We have received so many constructive suggestions from the five reviewers, and we feel that the revisions based on those suggestions have made the content easier to understand. Thank you very much once again.

  1. It is not clear why the authors have tried to tie all these information ranging from Statin mode of action to response to treatment, etc. This has made the manuscript to some extent compilation of data rather than a core-oriented work with a clear aim. This needs to be explained and the paper should be revised (especially the introduction) accordingly. 

We appreciate the reviewer’s comment. Our aim is to apply the results of our work to date, represented by this manuscript, to the implementation of the clinical trial that we have already prepared. To conduct an efficient clinical trial, it was necessary to establish biomarkers to select patients who would respond to statin, to confirm that statin would respond to clinical samples, to ensure the safety of administration, and to examine the synergistic effects of combination therapy to be tested in the clinical trial. We have reorganized the manuscript to clearly state our aims.

  1. Some tables can be removed and added to the supplementary material instead. 

We agree with the reviewer’s comment. Following a kind suggestion from other reviewers, we have moved tables 1 and 2 into the supplementary material in the revised manuscript.

Round 2

Reviewer 3 Report

Although authors have introduced some new experiments (such those in Fig 3B and 3C) that improved the manuscript, there is still the lack of a formal demonstration of a direct involvement of the identified genes in response to statin. This reviewer is still convinced of the need of experiments of gene modulation, silencing or over-expression, in the appropriated cellular models.

The manuscript is potential interesting but in the way it is presented is still only speculative, without mechanistic explanation.

Author Response

Response to Reviewer #3

Although authors have introduced some new experiments (such those in Fig 3B and 3C) that improved the manuscript, there is still the lack of a formal demonstration of a direct involvement of the identified genes in response to statin. This reviewer is still convinced of the need of experiments of gene modulation, silencing or over-expression, in the appropriated cellular models.

The manuscript is potential interesting but in the way it is presented is still only speculative, without mechanistic explanation.

We appreciate the reviewer’s comment. As the other 4 reviewers have already agreed, the aim of this manuscript is to search for conditions to design a clinical trial to confirm the efficacy of statins in ovarian cancer. By incorporating many constructive suggestions from reviewers, including you, we achieved to establish biomarkers to select patients who would respond to statin, to confirm that statin would respond to clinical samples, to ensure the safety of administration, and to examine the synergistic effects of combination therapy to be tested in clinical trial. The clarification of the mechanism by which VDAC1 and LDLRAP1 are involved in the response to statins is very interesting, and although it is somewhat far from the aim of this manuscript, it is expected in the future, thus we have added a note to it in the discussion (line 312-315).

Reviewer 4 Report

  1. About the mRNA expression in Figure 3a, is it compared with control? Or it has been normalized? How was it normalized (which gene was used to normalize)? Please add these information to Figure 3a.
  2. Since the synergistic effect is important in this work, the authors may need to explain in detail how combination index was calculated considering it can predict the synergistic effect.
  3. The authors need to explain how the isobologram can explain the synergistic effects, because not every reader is familiar with isobologram. Just try to make a reader who does not know isobologram to understand how to read Figure 5. For instance, how was the blue line acheived? What does the position of the points mean? In which area the points show suggests a good synergistic effect? Besides, is “1:1 and 4:1” in line 224 correct?

Author Response

Response to Reviewer #4

  1. About the mRNA expression in Figure 3a, is it compared with control? Or it has been normalized? How was it normalized (which gene was used to normalize)? Please add these information to Figure 3a.

We appreciate the reviewer’s comment. mRNA expressions were calculated by normalization of fluorescence intensity of VDAC1 or LDLRAP1 to those of the reference gene ACTB. This description has been added to the figure legends of Figure3a (line 171-173).

  1. Since the synergistic effect is important in this work, the authors may need to explain in detail how combination index was calculated considering it can predict the synergistic effect.

We appreciate and agree the reviewer’s comment. The method for calculating the combination index has been added to Methods section 4.9. in the revised manuscript (line 463-467).

  1. The authors need to explain how the isobologram can explain the synergistic effects, because not every reader is familiar with isobologram. Just try to make a reader who does not know isobologram to understand how to read Figure 5. For instance, how was the blue line acheived? What does the position of the points mean? In which area the points show suggests a good synergistic effect? Besides, is “1:1 and 4:1” in line 224 correct?

We appreciate and agree reviewer for the constructive comment. To make Figure 5 easier for the reader to understand, we have added what isobologram is, and what each point and connected blue line means in terms of synergistic effect in the figure legends of Figure5 (line 241-248). In addition, we would like to express our deepest gratitude to you for pointing out the misdescription of line 224. The misdescription has been corrected in the revised manuscript (line 227).

Reviewer 5 Report

The authors have addressed my comments, however the language still needs polishing by a native. 

Author Response

Response to Reviewer #5

The authors have addressed my comments, however the language still needs polishing by a native.

We appreciate the reviewer’s comment. We have asked a professional English editing company to proofread our manuscript, and the certificate is attached. The proofreader pointed out that the term “statins” is always referred to in the plural form, thus we have made this change throughout the text. In addition, we have incorporated all the suggested proofreading points into the text.

Round 3

Reviewer 3 Report

This manuscript addresses a very import topic that is drug repurposing. In particular authors suggest the use of statin for ovarian cancer treatment with a particular indication for clear cells and serous histotypes. Furthermore they suggest the use of two genes as predictive biomarkers of response, VDAC as associated to statin-response and LDLRAP1 as associated to statin resistance. By mean of combination assay they also suggest the use of statin together with some other available drugs and on the bases of treatment’s safety in mice they suggest the possibility to design clinical trial on selected population of patients.

If, as stated by authors: “the aim of this manuscript is to search for conditions to design a clinical trial to confirm the efficacy of statins in ovarian cancer. […] to select patients who would respond to statin, to confirm that statin would respond to clinical samples, to ensure the safety of administration, and to examine the synergistic effects of combination therapy to be tested in clinical trial”, the results should be stronger while they show only correlative data.

The selection of clear cell ovarian cancer as possible responding target is obtained only on long term cell lines, the cells lines obtained from ascites are not hystologically characterized. Data from HDRA are only cumulative.

The real involvement of VDAC and LDLRAP1 in statin sensitivity ad resistance is not demonstrated by experiments of silencing/overexpression.

Combination Indexes between simvastatin and other compounds are obtained on one cell line only.

Lines 155, 156 should be corrected, in the way the sentence is written, it looks like cell lines 4C and 7C are serous or clear cells and cell lines 2C and 8C are mucinous. However, accordingly to authors’ former response about characterization of these cell lines, their histological origin is unknown.

Author Response

Response to Reviewer #3

Lines 155, 156 should be corrected, in the way the sentence is written, it looks like cell lines 4C and 7C are serous or clear cells and cell lines 2C and 8C are mucinous. However, accordingly to authors’ former response about characterization of these cell lines, their histological origin is unknown.

We appreciate the reviewer’s comment. We have corrected and modified the description in the revised manuscript (line 152-157).